# The Immuno-Oncology and Genomic Aspects of DNA-Hypomethylating Therapeutics in Acute Myeloid Leukemia

**DOI:** 10.3390/ijms24043727

**Published:** 2023-02-13

**Authors:** Akiko Urabe, SungGi Chi, Yosuke Minami

**Affiliations:** Department of Hematology, National Cancer Center Hospital East, Kashiwa 277-8577, Chiba, Japan

**Keywords:** hypomethylating agents, acute myeloid leukemia, PD-1, CD47

## Abstract

Hypomethylating agents (HMAs) have been used for decades in the treatment of hematologic neoplasms, and now, have gathered attention again in terms of their combination with potent molecular-targeted agents such as a BCL-6 inhibitor venetoclax and an *IDH1* inhibitor ivosidenib, as well as a novel immune-checkpoint inhibitor (anit-CD47 antibody) megrolimab. Several studies have shown that leukemic cells have a distinct immunological microenvironment, which is at least partially due to genetic alterations such as the *TP53* mutation and epigenetic dysregulation. HMAs possibly improve intrinsic anti-leukemic immunity and sensitivity to immune therapies such as PD-1/PD-L1 inhibitors and anti-CD47 agents. This review describes the immuno-oncological backgrounds of the leukemic microenvironment and the therapeutic mechanisms of HMAs, as well as current clinical trials of HMAs and/or venetoclax-based combination therapies.

## 1. Introduction

Acute myeloid leukemia (AML) is an aggressive, rapidly progressing disease with a poor prognosis. Normal hematopoiesis is often disrupted in patients with AML due to the expansion of leukemic cells within the bone marrow. Although the morbidity rate of AML is higher in the elderly, its treatment can be challenging due to moderate tolerability and comorbidity. Intensive chemotherapy is generally provided to young adults without significant complications. Elderly patients aged 60 years and over have had historically poor performances when they were treated with intensive chemotherapy, with high reported early mortality rates of up to 40% [1]. Even in cases of a response, the duration of the response is limited, with a median overall survival (OS) of from 6 to 8 months. A variety of host- and disease-related risk factors for poor prognosis, including a history of myelodysplastic syndromes (MDS), unfavorable karyotypes, a poor performance status, and complications, limit the treatment options. As a result, older patients are likely to receive only palliative care. The National Comprehensive Cancer Network (NCCN) guidelines recommend standard induction chemotherapy (IC) for AML patients aged 60 years and older with a more favorable prognosis. Older patients eligible for IC may have relatively high complete remission (CR) rates. However, it has not been established that IC provides a clearer survival benefit compared with those of less-intensive treatment options. In a phase III trial, azacitidine (AZA) monotherapy showed better trends toward a favorable prognosis compared with those of conventional care regimens in older patients with newly diagnosed AML, with a median survival of 10 months [2]. In addition, a BCL-2 inhibitor venetoclax (VEN) showed superior survival benefits when it was in combination with AZA or LDAC, with a median OS of up to 14 months [3,4]. NCCN treatment recommendations for older patients with newly diagnosed AML also include decitabine, a hypomethylating agent (HMA). This review focused on immune-biological mechanisms of the AZA/VEN treatment, which have brought the paradigm shift of AML therapy. We also summarize the current status of the preclinical and clinical development of epigenetic immunotherapies.

## 2. Current Clinical Application of HMAs for AML

### 2.1. Combination of VEN and AZA in the Elderly

According to a phase III trial (VIALE-A) led by the University of Texas MD Anderson Cancer Centre, the combination of VEN and AZA was safe and improved the OS more than AZA alone did in selected patients with AML [3]. Patients treated with AZA plus VEN had a median OS of 14.7 months. In contrast, this was 9.6 months for patients in the AZA monotherapy arm. Furthermore, 66.4% of the patients in the AZA plus VEN combination reached CR, which was compared with 28.3% of the patients in the AZA monotherapy group. The responses to the treatments in both groups were rapid and sustained. Of the patients in the combination therapy group, 43% had a treatment responses in the first cycle, with a median duration of remission of 17.5 months. This result shows that the combination of VEN and AZA has a safety profile that is comparable to that of the two drugs administered separately. The most common adverse events in both the combination therapy and AZA plus placebo groups were hematological and gastrointestinal symptoms. Overall, the incidence of adverse events was consistent between the two treatment groups, although neutropenia (42% vs. 29%) and febrile neutropenia (42% vs. 19%) were more frequent in the combination therapy group with VEN and AZA than they were the AZA and the placebo group. That is to say, the toxicity profile of the VEN and AZA therapy was acceptable, with early mortality rates of less than 10%, despite an increased risk of myelosuppression and febrile neutropenia of up to 40%. More interestingly, the combination therapy with VEN and AZA was able to induce faster and more sustained remissions, with a median time to response of one month. This new regimen was approved by the FDA for newly diagnosed AML patients older than 75 years or those unfit for intensive chemotherapy.

Another type of AML that is more difficult to treat is treated secondary AML (ts-AML). This is a subtype of AML that develops following treatment for a precursor stage hematological disease such as myelodysplastic syndrome (MDS). Ts-AML is a poor prognosis disease as it does not tend to respond to treatment and has a high risk of early relapse. Despite the availability of approved therapies, patients previously treated with HMAs do not have improved outcomes, which was confirmed by a retrospective analysis of 562 patients with ts-AML with a history of HMA exposure who were treated with different types of treatments (IC, low-intensity chemotherapy (LIC), HAM, and VEN) [5]. The remission rates were investigated separately. This analysis showed that the combination of HAM and VEN resulted in higher remission rates and longer survival rates compared to those of IC and LIC. IC and LIC had similar CR (24% and 26% for IC and LIC, respectively) and OS rates (1 year survival: 14% and 22% for IC and LIC, respectively), while HAM and VEN therapies resulted in similar CR (39%) and OS rates (1 year survival: 35%). These results suggest that the combination of HAM and VEN may be the best option for treating patients with ts-AML with a history of HMA exposure.

### 2.2. Mechanism of Action of HMAs

In normal cells, DNA methyltransferase (DNMT)s function properly and the on/off balance of genes is maintained. However, in leukemic cells, the DNMTs do not work properly, resulting in the abnormal hypermethylation of DNA sequences that regulate the function of cancer-suppressor genes. As a result, many of the cancer-suppressor genes are switched off, making it easier for cancer to grow. AZA and decitabine are DNMT inhibitors.

They are analogues of the nucleoside cytidine and have two main anti-tumor activities: (i) cytotoxicity by incorporating into DNA or RNA and inducing a DNA damage response; (ii) DNA hypomethylation by inhibiting DNA methyltransferases, restoring normal growth and differentiation mechanisms. In addition, the molecular mechanisms of action of HMA are cellular uptake, intracellular activation, uptake into nucleic acids, and the induction of DNA hypomethylation through the inhibition of DNMTs (Figure 1). Cellular uptake is carried out by various nucleoside transporters; three types of phosphorylation produce the active metabolites 5-azacytidine-triphosphate for azacytidine and 5-aza-2’-deoxycytidine-triphosphate (5-aza-DCTP) for decitabine. HMA is replicated during incorporated into DNA and is, therefore, considered to be an S-phase-specific agent. During replication, 5-aza-dCTP from decitabine is incorporated into newly synthesized DNA. In contrast, 80–90% of AZA is incorporated into RNA as 5-aza-CTP, while 10–20% is multistep converted to 5-aza-dCTP by ribonucleotide reductase and incorporated into DNA.

### 2.3. Pathophysiologic Features of Leukemic Stem Cells

#### 2.3.1. Metabolic Changes in Leukemic Stem Cells

One study demonstrated that amino acid uptake, steady-state levels, and catabolism were all elevated in a leukemia stem cell (LSC) population. Furthermore, LSCs isolated from neonatal leukemia patients were found to be specifically dependent on amino acid metabolism for oxidative phosphorylation and survival. The pharmacological inhibition of amino acid metabolism reduces oxidative phosphorylation and induces cell death. In contrast, LSCs from patients with relapsed AML are independent of the amino acid metabolism, as they can compensate for this by increasing fatty acid metabolism. Amino acids are essential for oxidative phosphorylation in LSCs [6,7,8,9,10].

#### 2.3.2. Amplification of Anti-Apoptotic Factors

Cell death escape represents a cancer hallmark, and B-cell/CLL lymphoma 2 (BCL-2) family proteins play a pivotal role in tumorigenesis and survival [11]. Among these proteins, an anti-apoptotic factor myeloid leukemia-1 (MCL-1) is often highly amplified in a variety of human cancers [12]. MCL-1 strongly binds to pro-apoptotic effectors such as Bcl-2 homologous antagonist/killer (BAK) and the Bcl-2-interacting mediator of cell death (BIM) on the mitochondrial outer membrane, which results in the prevention of apoptosis [13]. Several studies have demonstrated that MCL-1 is essential for tumorigenesis and survival in solid tumors, lymphomas [14], and myeloid leukemia [15]. A recent multi-omics analysis of patient-derived LSC and a consequent xenograft mouse model experiment suggested that resistance to VEN-based therapy in monocytic AML relies on MCL-1 functions [16]. Potent MCL-1 inhibitors are currently under evaluation in clinical trials [17]. The combination therapy of AZA and MCL-1 inhibition may regulate fatty acid metabolism [18].

### 2.4. VEN and Gilteritinib

Relapsed/refractory *FLT3* mutant (*FLT3*+) AML is generally resistant to VEN, and a treatment with an *FLT3* inhibitor results in a shorter duration of the effects. In an open-label phase Ib study, 54 patients with R/R *FLT3*+ AML who were treated with VEN and the *FLT3* tyrosine kinase inhibitor, gilteritinib, participated [19]. The combination was safe and well tolerated, with 74.5% of the *FLT3*+ patients achieving a promising composite CR rate, except for frequent but manageable hemophilia, which occurred in 80% of the study participants. Furthermore, the combination therapy may contribute to prolonged survival, as more than half (56.7%) of the patients who responded to the treatment eliminated the FLT mutation. This trial presents promising evidence that VEN and gilteritinib may be effective and safe treatment options for patients with relapsed or refractory *FLT3*+AML. Further studies are warranted, including the development of a frontline combination of AZA and VEN plus gilteritinib in older AML patients and ineligible AML patients [20].

### 2.5. Magrolimab Combination Therapy

Patients with AML receiving a combination of VEN and AZA have high frontline remission rates, but they tend to eventually relapse, with very low survival rates in relapsed/refractory AML. A study focused on a treatment that adds magrolimab, an anti-CD47 antibody, to combination therapy [20]. Thirty-eight patients with various types of AML, comprising newly diagnosed AML patients, patients with relapsed/refractory AML who had not previously been treated with VEN, and patients with relapsed/refractory AML after a VEN treatment, participated in the ongoing, phase 2 study. The CR and CR with incomplete hematologic recovery (CRi) rates for newly diagnosed patients were 94%, which was 81% CR in the frontline patients. The CR/CRi rates in relapsed/refractory patients that were not treated with VEN were 63% and 27% in the patients with relapsed/refractory AML after the VEN treatment. Anemia occurring after the first and second doses is a common, but manageable side effect in the study participants, and it is a common side effect in VEN. The combination of AZA and magrolimab was found to be particularly effective in older patients with newly diagnosed AML and those unsuitable for IC. Furthermore, a randomized multinational phase III trial is now underway, and the results are promising (NCT04778397).

### 2.6. AZA and Ivosidenib, Newly Therapy

Approximately between 6% and 10% of AML patients have a disease induced by mutations in a gene called isocitrate dehydrogenase 1 (*IDH1*). Ivosidenib (IDB) inhibits the activity of a protein produced by *IDH1*. In a large international trial of patients with this type of AML who were unable to receive potent chemotherapy, IDB in combination with AZA was significantly more effective at inducing remission than AZA was alone [21]. Patients treated with both drugs in this clinical trial had a median OS of two years, an improvement compared with that of approximately 8 months due to AZA and a placebo. In general, the patients’ side effects were not worsened by receiving combination therapy. Indeed, the patients receiving combination therapy reported a better quality of life during the study compared with those of the patients receiving AZA monotherapy. The CR after 6 months was 11% in the AZA and placebo group compared with 38% in the combination group; beyond 6 months, further remissions were observed in those in the combination group. Overall, about half of the patients who received AZA and IDB achieved a CR compared with only 15% of those who received AZA and a placebo. Out of those who had a CR, more than half had no evidence of the *IDH1* mutation in their bone marrow. During the study period, 28 people (39%) died in the combination treatment group compared with 46 (62%) in the AZA and placebo group. Those receiving the combination reported that their health-related quality of life was the same or improved from that at the start of the trial. No improvement was reported in those receiving AZA and a placebo. Almost all of the study participants had one or more serious side effects due to the treatment, such as a decrease in red or white blood cells. Infections were most common in those receiving AZA and a placebo. Hemorrhage and differentiation syndromes (a series of dangerous complications attributable to the immune system) were more common in those who received AZA and IDB. These results suggest that the drug was more effective than the single agent was.

Thus, cancer patients with *IDH1* mutations can be treated with AZA and VEN or AZA and IDB. Both combinations, however, showed comparable life-prolonging effects to those of AZA alone in the randomized trials, but direct comparisons between the trials are not possible. In addition, clinical trials have already been initiated to test a combination therapy with AZA, VEN, and IDB. It would be desirable to further reduce the gap period required for the DNA sequencing of these leukemia cells. Trials comparing different targeted treatment options with the current standard of care based on the genetic profile of the patient’s tumor would also be necessary to determine which treatment is the best one.

## 3. Effect of HMAs on Tumor Immunity

### 3.1. HMA May Improve the Efficacy of CAR-T Cells

In an in vivo mouse models study, AZA induces the increased expression of CD123 in leukemia cells [22]. The administration of AZA to leukemic mice increases the number of CTLA-4-negative anti-CD123 CAR-T cells after the injection. These CAR-T cells show excellent cytotoxicity against AML cells, which is accompanied by increased production of TNFα and downstream phosphorylation of key T-cell-activating molecules. That is to say, AZA enhances the immunogenicity of AML cells and promotes the recognition and elimination of malignant cells by highly efficient CTLA-4-negative CD123 CART cells [23]. In other words, AZA enhances the immunogenicity of AML cells and promotes the recognition and elimination of malignant cells by high-efficiency CTLA-4-negative CD123CAR-T cells. The study also showed that anti-tumor activities, cytokine production, and proliferation are enhanced in decitabine-treated chimeric antigen receptor T (dCAR-T) cells both in vitro and in vivo. It suggests that HMA may improve the efficacy of CAR-T cells.

CD70 is a member of the tumor necrosis factor (TNF) superfamily and has recently been considered as a potential target for AML; the normal expression of CD70 is restricted to activated immune cells, but its expression is increased in many cancer types. Additionally, unlike other targets, CD70 is not expressed on normal hematopoietic stem cells, and an antibody targeting CD70, ARGX-110 (casatuzumab), showed excellent response rates in phase I trials. These results suggest that CAR-T therapies targeting CD70 may be effective. AZA is thought to increase CD70 surface expression in individual tumor cell lines and primary AML blast and leukemia stem cells. The mouse model also suggested that an increase in CD70 expression density would lead to a significant increase in CAR efficacy [24].

Chimeric antigen receptor T cell (CAR-T) immunotherapy targeting the CD19 antigen has shown clinical efficacy in patients with leukemia and lymphoma. However, it is less effective in lymphomas than it is in ALL. This is because CAR-T cells undergo a state of exhaustion that is characterized by the up-regulation of inhibitory receptors and loss of effector function [25,26,27,28,29]. It has recently been shown that de novo DNA methylation promotes T cell exhaustion and limits anti-PD1 immunotherapy and that methylation inhibition promotes PD1 blockade-mediated T cell rejuvenation [30,31,32,33]. Similarly, chromatin accessibility with functional DNA demethylation was found to be reduced in the transcription start regions in exhausted T cells.

The relapse of ALL is associated with persistence of CAR-T cells, suggesting that the active surveillance of leukemia via the CAR-T cells is lost. Previous studies suggested that decitabine (5-aza-deoxycytidine; DAC) may modify the DNA methylation program. In the present study, CAR-T cells after a low-dose DAC treatment (dCAR-T cells) underwent DNA initialization, induced a higher expression of genes for cell memory, proliferation, cytotoxicity and cytokine production, and reduced attrition following antigen exposure. Very low dCAR-T cell doses were able to efficiently control tumors with a very large tumor burden. Then, dCAR-T cells had a higher proportion of cells with a memory phenotype than CAR-T cells under long-term tumor stimulation. This means that the expression of memory-related genes was very high, and the expression of wasting-related genes was maintained at a low level. This suggests that the addition of DMAT may improve the attrition of CAR-T cells and increase their anti-tumor effect and reduce the tumor recurrence [34].

### 3.2. HMA-Induced Changes to the Immune System

HMAs are thought to induce an immunostimulatory effect in cancer cells through a process known as viral mimicry [35,36]. This effect is characterized by the up-regulation of repetitive transcription and formation of immunogenic double-stranded RNA (dsRNA) structures, leading to activity in the endogenous viral defense pathways and type I/III interferon responses. The high expression of viral defense gene signatures is also said to significantly correlate with durable responses to immune checkpoint inhibition (ICB) in melanoma patients, and synergistic effects have been reported where the combination of HMA and ICB results in enhanced immune infiltration and anti-tumor responses. The synergistic effect of combining HMA and ICB has been reported.

The intraperitoneal administration of DAC in immunocompromised mice was found to significantly inhibit tumor growth. These anti-tumor effects were quenched by the depletion of CD8+ T cells. A marked increase in CD8+ T cell infiltration into the tumor microenvironment was also observed during the DAC treatment; considering that a high density of CD8+ cytotoxic T cells is associated with prolonged disease-free survival, low-dose DAC was found to inhibit tumor growth by increasing CD8+ T cell infiltration and T cell effector function in vivo. DAC-activated CD8+ T cells were found to increase the protein expression of the T cell activation markers, CD69, CD25, and HLA-DR, which increased T cell activation and enhanced their ability to kill the target cells [37,38].

### 3.3. The Effect of TP53 Mutation on Anti-Leukemic Immunity

Patients with myelodysplastic syndrome (MDS) and secondary myeloid leukemia (sAML) with *TP53* mutations have a poor prognosis, with a median overall survival of 6–12 months regardless of the treatment. The aberrant expression of several immune checkpoints (e.g., PD-1, PDL-1, and CTLA4) in CD34-positive cells has been reported in 34% of MDS patients after an HMA treatment, which may be a poor prognostic factor. One study examined the relationship between checkpoint molecules and *TP53* gene mutations in MDS and sAML [39,40,41,42]. The up-regulation of the immune checkpoints PD-L1, PD-L2, and CTLA4 was found in 20–36% of the patients with *TP53* mutations compared to only 5–7% of wild-type *TP53* patients. Additionally, patients with the up-regulation of immune checkpoints such as PD-L1, PD-L2, and CTLA4 had an inferior OS rate (median OS 6 months vs. 19 months; *p* = 0.007). *TP53* mutant MDS/sAML also had increased HSCs and overexpressed PD-L1 (16.3% vs. 10.9%; *p* < 0.01).

The inducible T cell co-stimulator (ICOS) is a co-stimulator with the ability to induce tumor immunity by stimulating T cells, including cytotoxic T lymphocytes (CTLs) and Th cells, while promoting immune evasion by stimulating Treg cells. ICOS expression was markedly increased in the CTL and Th populations of *TP53* mutant patients relative to that of the wild type. Additionally, CTLs in the *TP53* mutant group were ICOS+/4-1BB/PD-1+, whereas Th cells were ICOS+ and Treg cells were low in ICOS+/PD-1. Increased numbers of ICOS-high/PD-1-negative Treg cells infiltrating tumors are associated with an inferior OS in cancer cells. Patients with increased ICOS-high/PD-1-negative Tregs in the bone marrow had a significantly reduced OS (median OS 8.6 months vs. 34.3 months; hazard ratio 2.34; *p* = 0.0006). This means that the number of OX40+ cytotoxic T cells (Treg) and helper T cells infiltrating the bone marrow was significantly reduced, especially in patients with TP53 mutations, as well as ICOS+ and 4-1BB+ natural killer cells. Furthermore, the number of immunosuppressive regulatory T cells (high levels of ICOS/PD-1-negative ones) and bone marrow-derived suppressor cells (PD1-low) is increased in *TP53* mutation cases [43]. The higher proportion of ICOS-high/PD-1-negative Treg cells infiltrating the bone marrow was highly advantageous as an independent prognostic predictor of overall survival. MYC was also significantly elevated in the *TP53* mutant and wild-type groups, and the PDL1-positive rate of MDS HSCs was positively correlated with the MYC expression (*p* = 0.004). MYC expression is negatively regulated by miR-34a, which degrades MYC mRNA and is induced by the wild-type p53 transcription target induced by wild-type p53. When we are comparing the differential expressions of miRNAs in bone marrow MPCs, miR-34a was significantly down-regulated in the *TP53* mutant group (*p* = 0.00003). In summary, PD-L1 expression was markedly increased in HSCs from the *TP53* mutant patients and was associated with up-regulation of MYC and the significant down-regulation of miR-34a, a negative regulator of MYC and the p53 transcriptional target.

In conclusion, the microenvironment of *TP53* mutant MDS and sAML has an immune-privileged, aversive phenotype that is a major contributor to a poor prognosis, suggesting that immunomodulatory treatment strategies may be effective in the future.

## 4. Impact of Genetic Backgrounds on DNA-Hypomethylating Therapeutics

### 4.1. Aberrant Epigenetic Modification in Hematologic Malignancies

#### 4.1.1. Physiologic Epigenetic Regulation of Hematopoiesis (Figure 2)

Hematopoietic stem cells (HSCs) continue self-renewal and generate several specialized progenitor cells to maintain normal hematopoiesis, which is mainly regulated by epigenetic modifications. In general, epigenetic modifications are classified into two major forms: the direct methylation of DNA and histone modification, such as methylation, phosphorylation, and acetylation. DNA methylation mainly occurs at the cytosine base that is linked to the adjacent guanine base through the phosphodiester bond (CpG). Although from two thirds to four fifths of CpGs are methylated through a whole genome, clusters of CpGs (so-called CpG islands) are often located on near promoter regions, and they are usually hypomethylated [44]. Unmethylated CpG island promoters show elevated levels of dimethylation of Lys4 of histone H3 (H3K4me2), suggesting that histone modifications contribute to the regulation of DNA methylation [45]. The genome-wide DNA methylation pattern is also associated with a variety of medical, psychological, and behavioral conditions [46,47,48,49], and these can be altered by extrinsic factors such as smoking [50] and coffee/tea consumption [51].

DNA methylation is carried out by DNA methyltransferase enzymes (DNMT) such as DNMT1, DNMT3A, DNMT3B, and DNMT3L. DNMT1 is involved in the methylation of existing GpC islands. A mouse model experiment demonstrated that the biallelic knockdown of *Dnmt1* led to a loss of the important function of embryonic stem cells [52]. DNMT3L does not have a catalytic function, and rather, it works as a co-factor of DNMT3A. DNMT3A proteins form a homo-tetramer or a hetero-tetramer with DNMT3L, which is important for its catalytic function [53]. DNMT3A and DNMT3B are involved in de novo DNA methylation and play a key role in the establishment of DNA methylation patterns during embryonic development [54,55].

High levels of DNA methylation of the promoter region are apparently associated with decreased transcriptional activity and gene silencing [44,56]. However, many key promoters of the regulator genes for embryonal development, which are constantly inactivated in mature cells, frequently have a large hypomethylated region named the DNA methylation valley (DMV) or canyon [57,58]. A large portion of DMVs are marked by the tri-methylation of the 27th lysine residue of histone H3 (H3K27me3), which is mediated by Polycomb repressive complex 2 (PRC2) [59]. PRC2 and PRC1 are polycomb group (PcG) proteins, which are evolutionarily conserved chromatin modifying factors. PcG proteins are known to play an important role in maintaining the stemness of murine embryonic stem cells by silencing key developmental regulators [60,61]. In addition to a pivotal role in embryonic development, PcG proteins also regulate the key function of adult hematopoietic stem cells (HSPs) [60,62]. A mouse model study demonstrated that a defective PRC2 assembly resulted in a decreased number of bone marrow HSPs and suppressed differentiation to mature cells [62]. The deposition of H3K27me3 is directly mediated by the enhancer of zeste homologs 1 (EZH1) and 2 (EZH2), a catalytic core subunit of PRC2, which are stimulated by other PRC2 subunits such as Jumonji and the AT-rich interaction domain containing 2 (JARID2) [63] and inhibited by EZH inhibitory protein (EZHIP) [64]. EZH2 also regulates the DNA methylation of EZH2-target promoters by directly binding with DNMT proteins [65]. 

PcG proteins keep DMVs hypomethylated in concert with ten–eleven translocation proteins (*TET*) [59]. *TET* proteins (*TET1*, *TET2*, and *TET3*) have a compact catalytic domain containing zinc fingers, and they specifically recognize CpGs. *TET* proteins are an alpha-ketoglutarate (aKG)-dependent dioxygenase that catalyze the conversion of 5-methylcytosine (5mC) into 5-hydroxymethylcytosine (5hmC), which is an initiation of DNA demethylation [66]. A genome-wide analysis of mice showed that the loss of *TET* genes led to increased methylation in DMVs, suggesting that *TET* proteins are essential in regulating the methylation of DMVs [59]. Recent studies have suggested that vitamin C, a co-factor of *TET* proteins, negatively regulates HSC function and prevent leukemogenesis, at least partially due to enhanced *TET* activity [67,68].

Additional sex comb-like 1 (*ASXL1*) protein was originally recognized as a co-factor of retinoic acid receptor [69]. *ASXL1* is a regulatory component of the Polycomb repressive deubiquitinase (PR-DUB) complex, which removes a mono-ubiquitin conjugated to the 119th lysine residue of histone H2A (H2A-K119Ub) [70,71]. The mono-ubiquitination of histone H2A is catalyzed by PRC1 [72], and it is essential for Polycomb-mediated transcriptional repression [73]. *ASXL1* also recognize N6-methyladenosine methylation (6mA) on DNA, which leads to its ubiquitination and the degradation by thyroid hormone receptor interactor 12 (TRIP12), and it is detracted by the deposition of 6mA [74].

**Figure 2 ijms-24-03727-f002:**
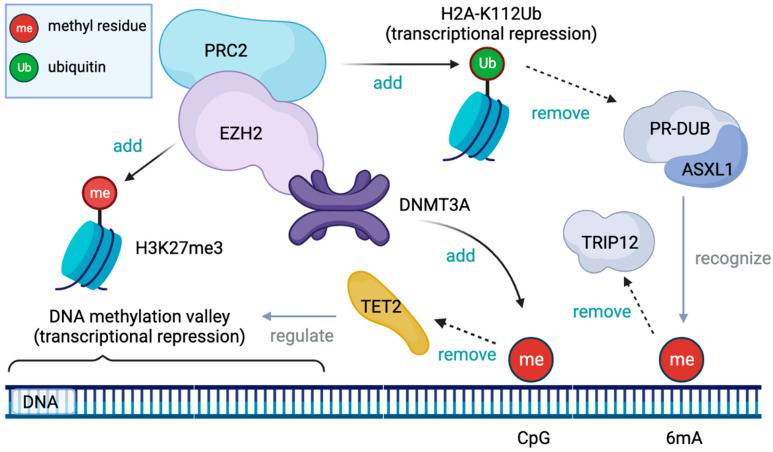
DNA methyltransferase 3A (DNMT3A) proteins methylate cytidine residue linked to the adjacent guanine residue (CpG). Ten-eleven translocation 2 (*TET2*) protein initiates the demethylation of CpG and contributes to maintaining the DNA methylation valley (DMV). Enhancer of zeste homolog 2 (EZH2) is a catalytic subunit of Polycomb repressive complex 2 (PRC2) and mediates the tri-methylation of the 27th lysine residue of histone H3 (H3K27me3), an epigenetic mark associated with DMVs. PRC2 also mediates the mono-ubiquitination of the 112th lysine residue of histone H2A, a mark of transcriptional repression, which is reversed by the Polycomb repressive deubiquitinase (PR-DUB) complex involving additional sex comb-like 1 (*ASXL1*) as its regulatory subunit. *ASXL1* also recognizes methylated adenine (6mA) on DNA and helps its degradation by thyroid hormone receptor interactor 12 (TRIP12).

#### 4.1.2. DNMT3A Mutation

DNMT3A mutations are relatively common in patients with cytogenetically normal acute myeloid leukemia (AML) and myelodysplastic syndrome (MDS) [75], accounting for a quarter of AML cases. The majority of the DNMT3A mutations are heterozygous missense mutations located in the 882nd arginine, though a broad range of pathogenic mutation sites have been reported. Because DNMT proteins function as an oligomer, most heterozygous loss-of-function mutations can disrupt part of its methyltransferase activity [76]. A DNMT3A mutation in AML is linked to decreased DNA methylation and a poor prognosis [77]. An analysis of the DNA methylation profiles in patients with cytogenetically normal AML revealed that DNMT3A mutations are associated with the hypomethylation of genomic regions far from the CpG islands (open see area) and specific promoter regions with a high CpG content such as homeobox (HOX) genes [78]. A mouse model experiment showed that concomitant homozygous *Dnmt3a* loss and the isocitrate dehydrogenase 2 (*IDH2*) gene mutation induced murine leukemia, accompanying epigenomic dysregulation such as the substantially decreased trimethylation (H3K9me3) and acetylation (H3K9ac) of the 9th lysine residue of histone H3, which was reversed by a histone deacetylase (HDAC) inhibitor [79]. A clinical retrospective analysis of patients with MDS who were treated with azacytidine and/or decitabine showed that the DNMT3A mutation was associated with a significantly better progression-free survival rate [80]. A similar retrospective analysis of patients with MDS or AML who received azacytidine therapy suggested that the DNMT3A mutation was associated with better response rates only if *TET2* and *IDH1/2* mutations co-existed [81].

#### 4.1.3. *TET2* Mutation

The *TET2* mutation is also commonly found in adult AML and MDS, accounting for 10–20% of patients [82,83]. A mouse model in vivo study suggested that the haploinsufficiency of *TET2* gene contributes to increased self-renewal of hematopoietic stem cells [84]. Decreased levels of 5hmC, which is catalyzed by *TET* enzymes, in genomic DNA in patients with myeloid malignancies have been reported, and low 5hmC levels are linked to hypomethylation at differentially methylated CpG sites [85]. The catalytic function of the *TET2* enzyme is also impaired by *IDH1/2* mutations [86]. Mutant IDH enzymes convert aKG into an onco-metabolite 2-hydroxyglutarate (2HG), and 2HG competitively inhibits aKG-dependent dioxygenases including *TET2* [87]. In clinical observations, *TET2* mutation in patients with AML or MDS may predict a better response to hypomethylating agents (HMAs) such as azacytidine [88,89]. An in vivo study using transplanted mice with leukemic cells carrying the homozygous *TET2* loss with the *Flt3*^ITD^ mutation demonstrated that an azacytidine treatment resulted in the normalized hypermethylated state of leukemic genome, as well as nearly the complete elimination of leukemic blasts, suggesting that *TET2*-mutated AML may be specifically responsive to HMAs [90]. A similar, positive predictive value for *TET2* mutation was reported in a retrospective analysis of patients with azacytidine-sensitive MDS [91].

#### 4.1.4. *ASXL1* Mutation

*ASXL1* mutations are prevalent in myeloid malignancies and associated with a poor prognosis especially when they have high variant allele fraction [92,93]. A transgenic mouse model study demonstrated that a gain-of-function mutation of the *ASXL1* gene resulted in an MDS-like condition in the mice, suggesting a molecular interaction with bromodomain containing 4 (BRD4) [94]. In a xenograft mouse model study in which human bone marrow cells from patients with chronic myelomonocytic leukemia (CMML) were transplanted into mice, the co-expression of mutant *ASXL1* and RUNX family transcription factor 1 (*RUNX1*) genes showed strong leukemic transformation, accompanying an increased expression of inhibitor of DNA binding 1 (*ID1*) gene [95]. Similarly, a knock-in mouse model study showed that co-existing *ASXL1*-G643W and CCAAT enhancer binding protein alpha (CEBPA) mutations synergistically accelerated leukemogenesis [96]. The clinical impacts of the *ASXL1* mutation on HMA therapy for patients with myeloid malignancies are still conflicting [80,97]. However, a retrospective analysis of patients with high-risk MDS who were treated with azacytidine showed a longer survival period for patients with the *ASXL1* mutation [97].

#### 4.1.5. EZH2 Mutation

EZH2 mutations are found in up to 10% of myeloid malignancies, including some AML cases [98]. Although *EZH2* gene is often overexpressed in many solid tumors, as well as malignant lymphoma, a defective EZH2 function should be important in myeloid malignancies [99]. A xenograft model study of MDS demonstrated that EZH2 knockdown resulted in reduced levels of H3K27me3 and the consequent up-regulation of HOX genes, which can contribute to the transformation to AML [100]. Similarly, in a mouse model study of MDS, the loss of EZH2 and concurrent *TET2* insufficiency induced the marked hypermethylation of CpG islands of the PcG target, which is characteristic of MDS [101]. Although the loss-of-function EZH2 mutation is associated with a shorter survival in patients with MDS [91], its prognostic impacts on AML patients are somewhat conflicting. A genomic analysis of 200 patients with AML revealed a poor prognosis after the first complete remission in patients with an EZH2-mutated intermediate-risk disease [102]. However, a larger cohort study (*n* = 1604) reported a comparable overall survival between patients with or without the EZH2 mutation [98]. Although a reduction of EZH2 levels and H3K27me3 can be achieved by DNA demethylation agents in combination with histone deacetylation inhibitors [99,103], it remains to be proved if such a therapeutic approach can be effective for EZH2-defective myeloid malignancies. Given that loss-of-function EZH2 mutation leads to hypermethylation of specific CpG regions [101], the therapeutic role of HMAs may not be ignorable.

### 4.2. Association between Anti-Leukemic Immunity and Epigenetic Dysregulation

Dufva and colleagues comprehensively analyzed the immunologic landscape of hematologic malignancies using 7092 samples from patients with hematologic malignancies, including 4281 cases of leukemia [104]. They demonstrated that 8.2% of the samples of AML showed high cytolytic activity, as well as the correlated infiltration of cytotoxic T lymphocytes into the bone marrow. Not surprisingly, the *TP53* mutation and deletion of chromosome 5p were positively correlated with high cytolytic activity, which may be due to a high mutational burden and more chance to generate neoantigens. In contrast, *FLT3* and nucleophosmin 1 (*NPM1*) mutations were associated with low cytolytic activity. AML samples with MDS-like transcriptome signature were related to high cytolytic activity and were also associated with MDS-related mutations such as *RUNX1*, *TP53*, *U2AF1*, and *SRSF2*. On the other hand, the expression of HLA class II and its transactivator (CIITA) was particularly low in samples of acute promyelocytic leukemia and *NPM1*-mutated AML. The methylation of promoter regions of HLA class II genes and *CIITA* was negatively correlated with cytolytic activity, and, of note, abundant *CIITA* hypermethylation was observed in subclusters with *IHD1*, *IHD2*, and *TET2* mutations. Interestingly, a subsequent cell line experiment showed that a co-culture with decitabine resulted in the decreased methylation of *CIITA* and increased expression of HLA class II. Given that HLA class II is essential for antigen presentation to CD4-positive helper T cells and the subsequent initiation of tumor-specific immunity, these findings indicate that a treatment with HMAs may contribute to enhance intrinsic anti-leukemic immunity and improve the response to immune therapies.

## 5. Conclusions

This review described the current development of AZA and/or VEN-based combination therapy including potent *FLT3* inhibitor gilteritib and novel immune-checkpoint inhibitor magrolimab, which have shown promising results in clinical trials. These approaches should bring benefits especially for patients who are ineligible for highly intensive chemotherapy and/or allogeneic hematopoietic stem cell transplantation. On an immune-oncological note, HMAs affect the leukemic microenvironment and possibly sensitize LSCs to immune-oncologic therapy such as immune checkpoint inhibitors and CAR-T cells. In addition, HMAs possibly reverse the immune-escaping mechanisms mediated by epigenetic dysregulation. HMAs are expected to play a key role in immuno-oncologic strategies for AML.

## Figures and Tables

**Figure 1 ijms-24-03727-f001:**
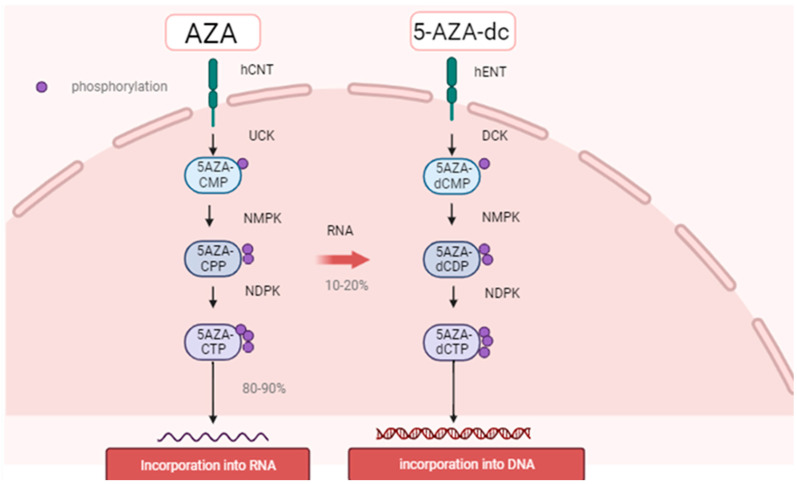
The figure shows 5-aza-U 5-aza-uridine, 5-aza-dU 5-aza-2’-deoxyuridine, CDP cytidine diphosphate, CMP cytidine monophosphate, hCNT human concentrative nucleoside transporter, hENT human equilibrative nucleoside transporter, NDPK nucleoside diphosphate kinase, NMPK nucleoside monophosphate kinase, and RNR ribonucleotide reductase, and azanucleoside (AZA) uptake and intracellular metabolism. Once inside the cell, the drugs are activated through consecutive ATP-dependent phosphorylation steps.

## Data Availability

Data sharing not applicable.

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
