# Peer review of "The Immuno-Oncology and Genomic Aspects of DNA-Hypomethylating Therapeutics in Acute Myeloid Leukemia"

_ijms, 2023, doi:10.3390/ijms24043727_

Round 1
Reviewer 1 Report
This review has been written to describe the immunobiology of hypomethylating agents in myeloid malignancies. It covers a vast topic and I suggest the following changes before its acceptable for publication
1. Section 2.1: please add long term follow up data from VIALE-A trial presented at ASH 2022 Pratz et al (paper #219)
2. Flow of manuscript needs to be worked on – under section 2 author starts by talking about combination therapy and switches to mechanism of action, LSCs biology and back to therapy. Suggest keeping therapy eg section 2.1,2.4, 2.5 ,2.6 etc under one subheading and separate basic biology under different subheading
3. Since review focusses on AML, suggest removing section on ALL in 3.1
4. Under combination strategies using aza+ ven please add data on pivekimab (ASH 2022; Daver et al) and aza+ven + gilteritinib (Nicholas Short et al ASH 2022)
5. Section missing on role of maintenance HMA in non HCT eligible patients (NEJM 2020; Quazar trial)
6. Under section 2 : authors have missed out on data using HMA+ immune checkpoint inhibitors – given nature of this review article it needs to be discussed under separate section as has been done for CD47 mab therapy
7. Section 2.6 discussed combination with IDH inhibitors – initial studies (Stein et and and Dinardo et al) in r/r AML setting also needs to be discussed for full context. Also mention ongoing studies using Pan IDH inhibitors
8. Context needs to be given for somatic mutations discussed ASXL1/DNMT3A /EZH2 etc. Discussion of recent ELN classification that allows reader to understand implications of mutations is needed
9. Conclusion section needs to be revised and authors need to discuss how HMA therapy has been beneficial to patients with myeloid malignancies, and comment on future improvements especially mentioning promising triplet combinations .
10. Minor point: the review may benefit from scientific writer to closely look at grammar and syntax
Author Response
Reviewer1
We are appreciated for your precise comments. We revised this paper following your suggestion, as well as adding new sections and references. Following sentence written in blue responded to your comments one by one.
Comments and Suggestions for Authors
This review has been written to describe the immunobiology of hypomethylating agents in myeloid malignancies. It covers a vast topic and I suggest the following changes before its acceptable for publication
- Section 2.1: please add long term follow up data from VIALE-A trial presented at ASH 2022 Pratz et al (paper #219)
We added the reference in section 2.1.
- Flow of manuscript needs to be worked on – under section 2 author starts by talking about combination therapy and switches to mechanism of action, LSCs biology and back to therapy. Suggest keeping therapy eg section 2.1,2.4, 2.5 ,2.6 etc under one subheading and separate basic biology under different subheading
We unified therapy sections into the same section. A summary of current clinical trials associated with HMA was also added.
- Since review focusses on AML, suggest removing section on ALL in 3.1
We removed all descriptions about ALL.
- Under combination strategies using aza+ ven please add data on pivekimab (ASH 2022; Daver et al) and aza+ven + gilteritinib (Nicholas prevalently ecpShort et al ASH 2022)
We added two references you recommended.
- Section missing on role of maintenance HMA in non HCT eligible patients (NEJM 2020; Quazar trial)
We created an additional section mentioning HMA maintenance.
- Under section 2 : authors have missed out on data using HMA+ immune checkpoint inhibitors – given nature of this review article it needs to be discussed under separate section as has been done for CD47 mab therapy
We mentioned about HMAs+ICIs in a separated paragraph, in the same context of the anti-CD47 antibody.
- Section 2.6 discussed combination with IDH inhibitors – initial studies (Stein et and and Dinardo et al) in r/r AML setting also needs to be discussed for full context. Also mention ongoing studies using Pan IDH inhibitors
We added description of initial studies of the IDH1 inhibitor, as well as an IDH2 inhibitor and second-generation IDH inhibitors.
- Context needs to be given for somatic mutations discussed ASXL1/DNMT3A /EZH2 etc. Discussion of recent ELN classification that allows reader to understand implications of mutations is needed
We added a section mentioning the AML risk stratification associated with aberrant epigenetic modulation.
- Conclusion section needs to be revised and authors need to discuss how HMA therapy has been beneficial to patients with myeloid malignancies, and comment on future improvements especially mentioning promising triplet combinations.
We revised conclusion, mentioning prospection of HMA-based treatment.
- Minor point: the review may benefit from scientific writer to closely look at grammar and syntax
This paper underwent proofreading by a professional entity again.

Reviewer 2 Report
This review discusses the clinical application of hypomethylating agents e.g. Azacitidine and its potential benefits for immunotherapies in treating AML. Most studies that this review refers to are up to date.
Specific comments:
(1) A typo "...such as a BCL-6 inhibitor venetoclax" is found in the abstract. It should be a BCL-2 inhibitor.
(2) In 2.1, "the incidence of adverse events was consistent between the two treatment groups...", I believe this statement is incorrect if the two treatment groups you talk about are AZA+VEN vs AZA+placebo. Because the trial VIALE-A (NCT02993523) clearly shows that AZA+VEN has higher incident rates than AZA+placebo in most documented adverse events except pneumonia. However, as you mentioned, it is acceptable with low early mortality rates. The original paper says "The safety profile of azacitidine plus venetoclax was consistent with the known side-effect profiles of both agents, and adverse events were consistent with expectations for an older AML population; no differences between the two treatment groups with respect to quality-of-life measures were seen. " And I believe that's what you really want to express, if so please rephrase. Besides, this particular trial also reports the composite remission of AZA+VEN vs control in cohorts harboring IDH1/IDH2, FLT3, NPM1 and TP53 mutations which are quite significant. This should not be overlooked.
(3) It is preferred to add some references for 2.2. Besides, you would like to double-check the legend of Figure 1 as the bold which I guess is the title but inserted in the middle of the legend. Two kinases UCK and DCK are not elaborated in the legend. Also, is it "CPP" a typo in the left route?
(4) What is the point for 2.4 in which you discuss the combination of venetoclax and gilterinib as your review is focusing on hypomethylating agents? If you want to cover the therapeutic benefits of combing AZA with gilteritinib, the phase III study is already available out there (https://ashpublications.org/blood/article/140/17/1845/486088/Phase-3-trial-of-gilteritinib-plus-azacitidine-vs).
Author Response
Reviewer2
We are appreciated for your precise comments. We revised this paper following your suggestion, as well as adding new sections and references. Following sentence written in blue responded to your comments one by one.
Comments and Suggestions for Authors
This review discusses the clinical application of hypomethylating agents e.g. Azacitidine and its potential benefits for immunotherapies in treating AML. Most studies that this review refers to are up to date.
Specific comments:
- A typo "...such as a BCL-6 inhibitor venetoclax" is found in the abstract. It should be a BCL-2 inhibitor.
You’re right. We corrected it.
- In 2.1, "the incidence of adverse events was consistent between the two treatment groups...", I believe this statement is incorrect if the two treatment groups you talk about are AZA+VEN vs AZA+placebo. Because the trial VIALE-A (NCT02993523) clearly shows that AZA+VEN has higher incident rates than AZA+placebo in most documented adverse events except pneumonia. However, as you mentioned, it is acceptable with low early mortality rates. The original paper says "The safety profile of azacitidine plus venetoclax was consistent with the known side-effect profiles of both agents, and adverse events were consistent with expectations for an older AML population; no differences between the two treatment groups with respect to quality-of-life measures were seen. " And I believe that's what you really want to express, if so please rephrase. Besides, this particular trial also reports the composite remission of AZA+VEN vs control in cohorts harboring IDH1/IDH2, FLT3, NPM1 and TP53 mutations which are quite significant. This should not be overlooked.
We revised description about adverse events: mentioning high frequency of febrile neutropenia in AZA+VEN arm as well as rare AE-related discontinuation and comparable QOL measures in both arms. The implication of gene mutations was also added in the text.
- It is preferred to add some references for 2.2. Besides, you would like to double-check the legend of Figure 1 as the bold which I guess is the title but inserted in the middle of the legend. Two kinases UCK and DCK are not elaborated in the legend. Also, is it "CPP" a typo in the left route?
We added two references related the description. We clarified what UCK and DCK stand for. “CPP” was a typo, CDP is correct.
- What is the point for 2.4 in which you discuss the combination of venetoclax and gilterinib as your review is focusing on hypomethylating agents? If you want to cover the therapeutic benefits of combing AZA with gilteritinib, the phase III study is already available out there (https://ashpublications.org/blood/article/140/17/1845/486088/Phase-3-trial-of-gilteritinib-plus-azacitidine-vs).
We added description about the LACEWING trial and the AZA+VEN part was removed.
